# Propolis: Its Role and Efficacy in Human Health and Diseases

**DOI:** 10.3390/molecules27186120

**Published:** 2022-09-19

**Authors:** Nadzirah Zullkiflee, Hussein Taha, Anwar Usman

**Affiliations:** 1Department of Chemistry, Faculty of Science, Universiti Brunei Darussalam, Jalan Tungku Link, Gadong BE1410, Brunei; 2Environmental and Life Science, Faculty of Science, Universiti Brunei Darussalam, Jalan Tungku Link, Gadong BE1410, Brunei

**Keywords:** propolis, chronic diseases, biological activity, active compounds

## Abstract

With technological advancements in the medicinal and pharmaceutical industries, numerous research studies have focused on the propolis produced by stingless bees (*Meliponini* tribe) and *Apis mellifera* honeybees as alternative complementary medicines for the potential treatment of various acute and chronic diseases. Propolis can be found in tropical and subtropical forests throughout the world. The composition of phytochemical constituents in propolis varies depending on the bee species, geographical location, botanical source, and environmental conditions. Typically, propolis contains lipid, beeswax, essential oils, pollen, and organic components. The latter include flavonoids, phenolic compounds, polyphenols, terpenes, terpenoids, coumarins, steroids, amino acids, and aromatic acids. The biologically active constituents of propolis, which include countless organic compounds such as artepillin C, caffeic acid, caffeic acid phenethyl ester, apigenin, chrysin, galangin, kaempferol, luteolin, genistein, naringin, pinocembrin, coumaric acid, and quercetin, have a broad spectrum of biological and therapeutic properties such as antidiabetic, anti-inflammatory, antioxidant, anticancer, rheumatoid arthritis, chronic obstruct pulmonary disorders, cardiovascular diseases, respiratory tract-related diseases, gastrointestinal disorders, as well as neuroprotective, immunomodulatory, and immuno-inflammatory agents. Therefore, this review aims to provide a summary of recent studies on the role of propolis, its constituents, its biologically active compounds, and their efficacy in the medicinal and pharmaceutical treatment of chronic diseases.

## 1. Introduction

Natural products have been regarded as a rich source of bioactive compounds due to the remarkable diversity of chemicals found in nature and have great medicinal and therapeutic potential [1]. A large number of studies have been intensively focused on exploring chemical compounds in natural products and assessing their biological potential, especially for medical treatments against a variety of acute or chronic diseases [2]. Among the natural products in focus, propolis has been discovered to be a promising alternative due to its numerous beneficial effects. In a number of in vivo and in vitro studies, it has been highlighted that many symptoms of diseases could be lessened and eased upon treated with propolis.

Bees gather the resins and beeswax from various parts of plants, such as flowers and leaf buds, exudates, resins, gums, and mucilage, surrounding their hive and then enriched them with their β-glucosidase enzymatic saliva [3,4,5,6]. Propolis is a valuable bee resinous product produced by different types of bees including *Apis mellifera* honeybees and stingless bees from the *Meliponini* tribe. The propolis of each species is very diverse and has a variety of chemical compositions. The botanical plants within a radius of up to a few kilometers surrounding beehives have a huge impact on the composition and physiochemical content of propolis.

Bees use propolis as a sealant to prevent cracks and crevices and to protect their hive against outside invaders as well as to stabilize both the temperature and moisture, and to inhibit any bacterial and fungal build-up, inside the hive [7,8]. In many cultures, propolis has long been recognized for its medicinal and therapeutic properties and has been historically recorded by the ancient Egyptians, Greeks, and Romans [9]. In particular, propolis has been used for skin beautification, mummification, preservative, and antiseptic to wounds, abscesses, and tumors [9]. In medieval times, propolis was extensively used as a “herbal” medicine in Eastern Europe and in the Middle East. In the early modern era, propolis became a research subject, particularly with respect to the identification of its chemical compositions.

Propolis has been documented to have more than 500 compounds, including flavonoids, phenolic compounds, polyphenols, terpenes, terpenoids, coumarins, steroids, amino acids, and aromatic acids [3,10,11,12]. In addition, propolis is rich in phytochemicals, including essential oils, vitamins (A, B complexes, C, and E), and important minerals, such as aluminum, sodium, potassium, calcium, copper, magnesium, iron, and zinc [13,14], which also play important roles in terms of biological activity [15]. As bees collect the raw materials of propolis from various parts of plants, pigments and compounds produced by a variety of plants are found in propolis, and the chemical compositions vary depending on its geographical location, botanical sources, season, and bee species [16]. In the modern era, research on propolis has been directly based on a deeper understanding of its biological activity, and in numerous preclinical and clinical studies, it has been proven that a wide range of natural compounds, including flavonoids, phenolic compounds, polyphenols, terpenes, terpenoids, and aromatic acids in propolis are potential anticancer, antiapoptotic, antidiabetic, anti-inflammatory, antioxidant, antibacterial, and antiviral agents [17].

Recently, with the development and improvement in technology for medicines, there has been an emerging interest in the study of propolis and its diverse constituents. In addition to its medicinal and therapeutic properties to treat various chronic diseases, propolis has been effectively applied in the treatment of diabetes, burns, wounds, gynecological problems, laryngological, dermatological, and neurodegenerative diseases, gastrointestinal disease, respiratory tract-related diseases, and cardiovascular disorders, as well as COVID-19. The potential uses of propolis in human health and diseases are schematically shown in Figure 1.

Most studies have focused more on *A. mellifera* propolis than stingless bee propolis [3]. Recently, applications of stingless bee propolis in human health and chronic diseases have been intensively investigated, and they have been well summarized in a small number of review articles [6,7,8,9]. This current review article summarizes recently reported studies on *A. mellifera* and stingless bee propolis and its role and efficacy in the treatment of various acute and chronic diseases, its significant benefits, and its main biologically active compounds and their classifications. In addition, this article focuses on the vital roles of these biological compounds in the prevention and treatment of diseases.

## 2. Chemical Constituents

It is noteworthy that, typically, propolis contains mainly lipid (50%), beeswax (30%), essential oils (10%), pollen (5%), and organic components (5%). The lipid content of stingless bee propolis is several folds greater than that of *Apis melifera* honeybee hives (8–16%), so stingless bee propolis is more water-resistant than honeybee hives [15,18]. The lipids are collected by bees from plant resins [19,20,21]. In general, the organic components consist of carboxylic acids (20%), terpenoids (15%), steroids (12%), hydrocarbons (10%), sugars (6%), alkaloids (6%), flavonoids (4%), phenols (3%), vitamins (2%), amino acids (2%), ketones (2%), proteins (1%), and other compounds (14%) [22,23]. The most important biological active compounds are flavonoids, polyphenols, carboxylic acids, quercetins, fatty acids, cinnamic acid, esters, and terpenoids, such as pinocembrin, galangin, carboxylic acids, caffeic acid, caffeic acid phenethyl ester (CAPE), saponin, phorbol, naringenin, gallic acid, naringin, benzoic acids, amino acids, apigenin, coumaric acid, steroids, vitamins, reducing sugar, and essential oils [24,25], as summarized in Table 1.

Flavonoid and phenolic compounds are an important class of plant secondary metabolites. Among flavonoids, quercetin and chrysin are mostly found and distributed in the propolis of different bee species [4]. These flavonoids are the active compounds in plant resin, and they have various biological activities such as anticancer, antimicrobial, and anti-inflammatory activities [26,27,28]. The flavonoids are sub-categorized into several classes including flavone, flavanone, flavan, isoflavone, flavanol, flavanonol, flavan-3-ol, and chalcone. The classification of phenolic compounds can be seen in Figure 2. Phenolic compounds, for instance, quinones, benzophenones, coumarins, tannins, and lignans (see Figure 3) [29], naturally originate from fruits, vegetables, leaves, bark and roots. The relationship between the chemical constituents in propolis and their plant origins has been well documented by Bankova et al. [30], and it could be anticipated that propolis consisting of different chemical constituents has a variety of biological active compounds, bioactivities, and pharmaceutical activities. For examples, naturally phenolic compounds have important roles in the context of protecting plants from sunlight, herbivorous animals, and microbial pathogens. Similar to flavonoids, phenolic compounds also exhibit biological activities such as antioxidant, antibacterial, anticancer, anti-inflammatory, antitumor, plasmodicidal, and anti-HIV activities [31,32,33].

Limonene, α-cubebene, and β-caryophyllene are a select example of the most prevalent terpenes, while lupeol and β-amyrin are triterpenes identified in propolis [34]. These terpenes and triterpenes are classified as volatile biosynthetic compounds and originate from plants. Other volatile compounds present in propolis include esters of fatty acids and organic acids. In plants, all these volatile compounds play an important role in the context of attracting insects to aid the pollination process, and they are potentially applicable as spices, fragrances, and flavors in perfumery, cosmetics, pharmacies, and food products. In particular, the broad spectrum of the biological activities of such volatile compounds have been pointed out to have analgesic, anticancer, anti-inflammatory, antitumor, antifungal, antimicrobial, antiviral, and antiparasitic activities [35,36]. It has been demonstrated in many studies that the biological activity and quality of propolis are directly related to its chemical constituents [37]. Therefore, it is not surprising that the chemical constituents of propolis, in addition to its physicochemical properties, is used as a standardized parameter for its quality [38]. Most of the common bioactive compounds responsible for the biological and pharmaceutical properties of propolis are apigenin, chrysin, galangin, luteolin, kaempferol, pinobanksin, pinocembrin, quercetin, caffeic acid, cinnamic acid, p-coumaric acid, ferulic acid, artepillin C, CAPE, and coumarin, as summarized in Figure 4.

## 3. Pharmaceutical and Therapeutic Properties of Propolis

As mentioned in the previous section, several studies confirmed that propolis has therapeutic potential in pharmacies and medicines to treat various chronic diseases, particularly autoimmune diseases, diabetes, burns, wounds, gynecological problems, and laryngological, dermatological, neurodegenerative, gastrointestinal, and respiratory tract-related diseases, cardiovascular disorders, antimicrobial, anticancer and antioxidant activities, and COVID-19. Here, the pharmaceutical and therapeutic properties of propolis are summarized. Accordingly, the bioactive components of propolis related to their effects on various medical conditions and diseases are gathered in Table 2.

### 3.1. Autoimmune Diseases

#### 3.1.1. Diabetes Mellitus (Type 2)

Type 2 diabetes mellitus (T2DM) is a medical condition that arises due to an elevated level of blood sugar (glucose) that is caused by the body not being able to produce enough insulin [39]. T2DM has attracted great interest in the research of natural compounds as a means of prevention [39]. In this sense, a small number of studies have shown that propolis, which contains flavonoids that are active ingredients as antioxidant, anti-inflammatory, and free radical-scavenging agents, may have beneficial therapeutic values for the treatments of complex diseases, including T2DM [40,41]. Indeed, apigenin, chrysin, galangin, kaempferol, luteolin, genistein, and pinocembrin found in propolis have been identified to possess antidiabetic effects [34]. In addition, naringin, which is a natural flavanone glycoside found in propolis, has been reported to have insulin-like and lipid-reducing properties that reduce both insulin resistance and hyperglycemia, and this is in addition to its anti-inflammatory, antioxidant, anticancer, antiapoptotic, and anti-osteoporosis properties [34]. Both apigenin and naringin have been administered to increase the inhibition of enzyme glycogen phosphorylase and glucose uptake by muscle cells. These flavonoids, including apigenin, naringin, chrysin, galangin, kaempferol, luteolin, genistein, and quercetin help to reduce blood glucose concentration, to detect insulin in islets or serum, and to prevent insulin release [34].

Recently, a small number of studies have further demonstrated that the use of propolis showed significant effects, reducing the blood glucose levels, serum insulin, and serum glycosylated haemoglobin (HbA1c) levels of T2DM patients [40,42,43]. It has been suggested that propolis could influence glucose metabolism through the suppression of intestinal α-glucosidase activity in carbohydrate digestion and stimulate the β-cells in the islets of Langerhans in the pancreas, resulting in an increase in insulin secretion [35]. Zakerkish et al. pointed out that patients with T2DM who received propolis supplementation for 12 weeks had lower insulin levels and insulin resistance [40]. Various reports have also shown that the development and complications of T2DM are highly influenced by free radicals, oxidative stress, and inflammatory cytokines [43]. Active constituents in propolis can scavenge free radicals, lower blood glucose levels, and modulate blood lipid metabolism [36]. The oxidative stress which results in the production of reactive oxygen species (ROS) stimulates inflammation and inflammatory mediators [34]. T2DM associated with ROS then causes oxidative damage to important organs, such as the heart, kidneys, nerves, and eyes [34]. Subclinical inflammation, which leads to insulin resistance, is related to the characteristics of metabolic syndrome and hyperglycemia. High glucose levels deplete the absorption of intestinal carbohydrates and trigger glucose uptake by peripheral tissue [35]. In this sense, the antihyperglycemic effects of propolis inhibit the production of glucose from dietary carbohydrates, and propolis can regulate an increase in postprandial glucose and improve insulin resistance [35]. Propolis also improves the glycemic and lipid profiles of T2DM patients [42], demonstrating that propolis is a promising agent in preventing and controlling diabetes mellitus.

#### 3.1.2. Rheumatoid Arthritis

Rheumatoid arthritis (RA) is an immuno-inflammatory chronic disorder that leads to severe pain and functional limitations in the joints that are triggered by the immune system of the human body [29]. The pathogenesis and related outcomes of RA are significantly influenced by oxidative stress and inflammation. Oxidative stress decreases the blood antioxidants of patients with RA [29]. Inflammation is positively related to the development of RA and has been attributed to the activation of nuclear factor kappa B (NF-κB) by ROS [31]. There are several factors that may affect the immune response, resulting in the release of pro-inflammatory cytokines. These cytokines can cause joint regenerative changes that are caused by inflammation and synovial cell activation. Common cytokines include tumor necrosis factor-alpha (TNF-α), interleukin-1 beta (IL-1β), and interleukin-6 (IL-6).

The treatment of RA with natural supplements with fewer side effects and cost-effectiveness, such as propolis, has been confirmed to be beneficial due to their antioxidant and anti-inflammatory effects [32]. In particular, the chemical compounds of propolis, such as terpenoids, phenolics, steroids, alcohols, terpenes, and sugars, have been shown to be responsible for treatment. Additionally, propolis has been associated with the influence of bone, cartilage, and tooth pulp regeneration. Furthermore, propolis can also suppress inflammatory cascades by blocking the NF-κB pathway and reducing ROS by enhancing antioxidants [29]. This is because the affected joints of RA patients have increased oxidative stress and inflammation, which were reduced by chemical compounds [29]. In other words, the chemical compounds of propolis have powerful anti-inflammatory properties that allow them to regulate the basic function of immune cells and decrease the cytokines mediated by the immune response in T-cells and NF-κB activation [26,27]. Ansorge et al. highlighted that both DNA synthesis and inflammatory development in T cells can be suppressed, while the formation of the transforming growth factor-β1 (TGF-β1) of the cells are promoted by the caffeic acid, CAPE, hesperidin, and quercetin of propolis [28]. Zhang et al. reported that apigenin and galangin decrease the mRNA levels of TNF-α [37]. In addition, CAPE is a very significant compound of propolis that has anti-inflammatory properties and also acts as the selective inhibitor of NF-κB activation as well as being able to precisely inhibit NF-κB activation by extensive inflammatory stimuli including TNF-α [38].

ROS are formed during aerobic metabolism, and the antioxidant defense system protects the cell against ROS [44]. It is conceivable that when the amount of ROS produced is more than what the antioxidant system can handle, oxidative stress leads to metabolic failure and significant damage to DNA, lipids, and proteins [44]. In rheumatoid arthritis, the activation of macrophages leads to an increase in ROS, which are a very significant mediator in arthritis [44]. In addition, antioxidants could also inhibit the production of the cytokines that are induced by TNF-α, which is a protective mechanism against rheumatoid arthritis. Kurek-Górecka et al. reported that the antioxidant mechanism of polyphenols found in propolis may be attributed to their scavenging effects on ROS [39]. The production of free radicals and their synergistic effects with antioxidants are influenced by the chelation of metal ions and nitrogen species, which are useful in reducing RA [39].

Unfortunately, the adverse effects of propolis on the activity of RA disease are still uncertain. A multicenter, double-blind, randomization, monitored trial has been implemented in order to evaluate the effect of propolis in patients with RA [45]. The clinical trials revealed that stingless bee propolis did not improve RA quality of life or reduce disease activity. The lack of effect of propolis on the suppression of disease activity were associated with medications received by participants prior to the clinical study. In contrast, Brazilian propolis have been reported to have positive effects in reducing the activity of RA disease in mice, demonstrating that propolis may provide novel therapy options [45]. The phosphorylation of the signal transducer and activator of the transcription pathway was inhibited by propolis, leading to a reduction in the production of interleukin-17 that contributes to RA disease. A case study on the effects of propolis on the activity of RA disease has therefore been suggested, taking account of additional factors that might aid a human trial, such as the effects of propolis on the length of recovery time, the amount of doses, and an extension of the number of male and female participants [45].

#### 3.1.3. Anticancer

Cancer is defined as a disease caused by the abnormal growth of cells with the potential to spread and invade other parts of the body. Some cancer cells may be resistant to chemotherapy and the efficacy of certain drugs may also be reduced as a result of cancer cells developing drug resistance. According to the World Health Organization (WHO), one of the leading causes of mortality worldwide is cancer. So far, there have been no cures for cancer, but there are numerous treatments that can help to reduce the spread of tumors. Thus, due to the inefficacy of currently available medicines, scientific research has focused on the production of potential drugs from natural resources for cancer treatment. Nowadays, one of the most popular alternative drugs used is natural products. This is due to the wide range of natural bioactive compounds that exert beneficial effects on human health. It was also stated that most anticancer agents are derived from natural products [46].

The study of the anticancer properties of propolis resulting from different bee species found in various geographical locations in the treatment of breast, colon, liver, lung, and pancreatic cancer cell lines has been extensively documented in recent years [24,47,48]. Some studies focused on the efficacy of anticancer therapy by evaluating the ability of cancer cells to initiate apoptosis and cell cycle arrest, which are the main mechanisms of the anticancer properties of propolis [24,47,49]. The diversity of chemical compounds of propolis highly affects its anticancer activity. The active ingredients of propolis, such as flavonoids, exhibit chemopreventive effects against most carcinogenesis. Other active compounds of propolis with anticancer and antiproliferative properties include apigenin, caffeic acid, CAPE, ferulic acid, galangin, luteolin, myricetin, pinocembrin, and quercetin [50,51]. Propolis also acts as pro-apoptotic protein, activates the caspase cascade mechanisms, and releases cytochrome C from the mitochondria into the cytosol in order to target molecules that are important in apoptosis through the intrinsic pathway [39].

In addition, propolis has been shown to exhibit synergistic effects in radiation and chemotherapy medications for breast cancer [50]. Overall, propolis reduces proliferation, triggers apoptosis, prevents metastases, and inhibits the progression of the cell cycle [50]. Vatansever et al. showed that bioactive components such as caffeic acid and galangin in Turkish propolis have antiproliferative effects on breast cancer, colon cancer, and liver cancer cells, where compounds promote apoptosis and inhibit proliferation by reducing the viability of cancer cells [52]. The same results were observed for propolis from Indonesia, Greece, and Serbia [50,51], where luteolin and myricetin were exposed to breast cancer cells, with these compounds exhibiting consistent cytotoxic activity. Meanwhile, galangin is one of the most abundant flavonoids in propolis from Algeria and Brazil and plays an important role in controlling cell migration and cell adhesion as well as inducing lung cancer cells [53,54]. Caffeic acid and its derivatives found in Algerian propolis also present anticancer and antiproliferative activities [53,55]. It was also discovered that Chinese propolis enhances cell cycle arrest and causes apoptosis in pancreatic cancer cells [56]. In addition, the artepillin C, CAPE, galangin, kaempferol, and quercetin found in Chinese, Brazilian, Korean propolis exhibit strong antiangiogenic properties which help to prevent inflammation and cancer [57,58].

Ebeid et al. described a clinical test featuring propolis on a group of 135 patients diagnosed with breast cancer undergoing radiation therapy, where the participants were split into three groups and subjected to different tests in order to compare their specific reactivity on propolis supplementation based on age range, menopausal status, and radiotherapy [59]. A significant decrease in radiation-induced DNA damage was reported in concerning the group of patients who underwent radiotherapy and received propolis supplementation due to the ionizing radiation of leukocytes from breast cancer patients. This study therefore suggested that propolis helps to boost the efficiency of serum to neutralize free radicals as well as the effect the way in which iron is absorbed by the body and hemoglobin is produced [59]. Piredda et al. investigated the tolerability, safety, and adherence of propolis in breast cancer patients taking medications as well as the effect of propolis on preventing oral mucositis, and it was demonstrated that the combination of propolis and bicarbonate prevented oral mucositis in breast cancer patients [60]. Darvishi et al. conducted a clinical trial to compare the antioxidant and anti-inflammatory benefits of a propolis supplement to cancer patients receiving chemotherapy [61]. Patients in the placebo group significantly increased their levels of pro-inflammatory cytokine tumor necrosis factor, which is a biomarker of oxidative stress, while patients with propolis supplementation did not show a significant increase in their pro-inflammatory cytokines levels, but their pro-oxidant antioxidant balance was decreased.

### 3.2. Cardiovascular Disease

Cardiovascular disease (CVD) is one of the leading risk factors worldwide associated with heart and blood vessels and can cause heart attack, stroke, and angina [62]. It is normally caused by the narrowing or blocking of blood vessels [62]. Interestingly, it has been suggested that propolis has positive effects on the treatment of cardiovascular diseases such as hypertension, atherosclerosis, and ischemia-reperfusion (IR) injury [63]. The chances of CVDs being affected by risk factors such as oxidative stress and obesity are very high, but propolis supplementation, with its bioactive components, could reduce the risks associated with CVD [63].

It has been mentioned in the literature that the cardiovascular effects of propolis are frequently associated with its antioxidant, anti-inflammatory, immunomodulatory, antihypertensive, antiangiogenic, and antiatherosclerosis properties [52]. In this sense, propolis has a variety of cardioprotective properties due to its active ingredients, particularly its phenolic compounds, such as chrysin, luteolin, pinocembrin, and quercetin [57]. The phenolic compounds of propolis reduce the activity of cyclooxygenase, ROS, and nitric oxide (NO) productions, which are also related to the antioxidant properties of propolis [64]. CAPE, another major compound of propolis, also possesses an antioxidant property and exhibits protective effects against IR injury in different tissues, including the brain, colon, heart, and liver [57]. Ahmed et al. suggested that Malaysian propolis showed cardioprotective activity and antioxidant properties against isoproterenol-induced oxidative stress via cytotoxic radical-scavenging [65]. In addition, flavonoids of propolis also have the ability to prevent the progression of pathological cardiac hypertrophy and heart conditions [66].

### 3.3. Coronavirus Disease

In early 2020, as declared by the WHO, the world was affected by the global outbreak of an infectious disease resulting from severe acute respiratory syndrome coronavirus 2 (SARS-CoV-2), also known as the coronavirus disease (COVID-19) [67]. It is a very contagious disease that has resulted in widespread symptoms such as a cough, fever, body ache, a weakened immune system, acute respiratory problems, and ultimately mortality [68]. In addition to the recently discovered disease, the treatment for COVID-19 has been limited until recently. Given the rapid mutation and complexity of the COVID-19 virus, there has been an extensive effort to establish effective treatments against the disease. Since then, many researchers have focused their research on discovering alternative treatments that use natural resources, including bee products such as propolis [69,70,71,72,73,74]. This is because propolis contains biological properties that are applicable to the SARS-CoV-2 infection, such as anti-inflammatory effects, the boosting of the immune system, and reducing viral replications [70].

Some clinical studies have shown that the presence of propolis may alleviate the treatment and relieve the symptoms of COVID-19 patients [70,71,73]. This is because the potential active phytochemical constituents of propolis, which include alkaloids, flavonoids, lipids, phenols, polyphenols, saponins, and tannins, all of which are strongly associated with the pharmacological properties of propolis. Marcucci had stated that the antiviral activities of caffeic acids, flavonoids, aromatic acids, and their esters in propolis may prevent the transmission as well as the propagation of the virus [7]. By inducing apoptosis or by blocking cellular pathways, propolis and its active compounds may also reduce the viability of endothelial cells, which promote cell migration and proliferation [70]. Active ingredients found in propolis, such as artepillin C, baccharin, caffeic acid, drupanin, myricetin, and quercetin, display a sufficient amount of inhibition against SARS-CoV-2 helicase in human body [75].

Although several therapy options for COVID-19 have been investigated, and the best course of action has not yet been determined, several patients have been assigned to the treatment of propolis supplementation therapy, in which each patient has to consume 400 mg or 800 mg/day of propolis for a week [76]. It has been demonstrated that there is strong evidence that propolis is not only safe for consumption but that it can also be used to treat a number of disorders associated with severe cases of COVID-19. This is because propolis is highly associated with its active ingredients such as apigenin, artepillin C, quercetin, caffeic acid, CAPE, and kaempferol. In particular, quercetin possess a number of properties that are essential in the treatment of COVID-19 including antiviral, anticancer and senolytic activities [76]. Meanwhile, the other components mentioned above can prevent the oncogenic kinase PAK1, which stimulates a harmful immune response.

One of the potential defense mechanisms of propolis against SARS-CoV-2 infection is an interference with the viral entry and replication process [69,77]. An in vitro study documented the effect of flavonoids on DNA and RNA viruses, including COVID-19 [63]. From this study, both chrysin and kaempferol were exhibited to possess a significant activity in the inhibition of viral replication [78]. Further evidence showed that the interactions between the active constituents of propolis and SARS-CoV-2 resulted in the activity of RNA-dependent RNA polymerase inhibition and might also synergistically increase its antiviral properties [69]. Sahlan et al. demonstrated that angiotensin-converting enzyme 2 (ACE2), which is a receptor in the human body, has been recognized by SARS-CoV-2 [74]. SARS-CoV-2 major protease is a potential therapeutic target for COVID-19 [74]. Furthermore, Berretta et al. also pointed out that the interactions between propolis and ACE2 result in a reduction in the invasion of the host cell by SARS-CoV-2 [71]. The inhibition of the ACE2 enzyme is a significant target for the treatment of COVID-19. This is due to its involvement in the binding and membrane fusion process, which may be potential targets in the development of antiviral properties for COVID-19 treatment.

**Table 2 molecules-27-06120-t002:** Active components of propolis and their outcome effects on various medical conditions and diseases.

Medical Disorder	Propolis Active Components	Type of Diseases	Outcomes	Ref.
Auto-immune disease	Apigenin, chrysin, galangin, genistein, kaempferol, luteolin, naringin, pinocembrin, and quercetin	○ Diabetes mellitus (Type 2)	○ Reduces blood glucose levels ○ Decreases serum glycosylated haemoglobin (HbA1c) ○ Reduces insulin levels ○ Scavenges free radicals and chelate metals	[18,22]
	Apigenin, caffeic acid, CAPE, galangin, hesperidin, and quercetin	○ Rheumatoid arthritis	○ Diminishes inflammation ○ Suppresses DNA synthesis and the production of inflammatory cytokines ○ Inhibits NF-κB activation ○ Decreases the mRNA levels of TNF-α	[22,23,24,27,28]
Cancer	Apigenin, artepillin C, caffeic acid, CAPE, chrysin, galangin, kaempferol, luteolin, myricetin, pinocembrin, and quercetin	○ Breast cancer ○ Colon cancer ○ Liver cancer ○ Lung cancer ○ Pancreatic cancer	○ Prevents cell proliferation ○ Inhibits the angiogenesis of cancer cells ○ Stimulates apoptosis ○ Selective toxic properties against tumor cells ○ Apoptosis effects	[50,53,79,80]
Cardiovascular diseases	CAPE, chrysin, kaempferide, luteolin, pinocembrin, pinostrobin, and quercetin	○ Stroke ○ Hypertension ○ Atherosclerosis ○ Ischemia-reperfusion injury	○ Reduces the activity of cyclooxygenase ○ Scavenges reactive oxygen species (ROS) ○ Stimulates nitric oxide (NO) production	[57,63,64]
Severe acute respiratory syndrome coronavirus 2 (SARS-CoV-2)	Artepillin C, baccharin, caffeic acid, CAPE, drupanin, myricetin, and quercetin	○ Coronavirus disease (COVID-19)	○ Inhibits coronavirus in the human body ○ Prevents the viral transmission and viral propagation of the virus	[69,70,71,72,73,74,75]
Gastrointestinal disorder	Artepillin C, CAPE, coumaric acid, galangin, kaempferide, 4-methyl ester, and aromadendrin	○ Gastric ulcer ○ Peptic ulcer ○ Cancer ○ Oral mucositis ○ Gastritis colitis ○ Mucositis ulcers	○ Protects against gastric ulceritis ○ Improves the intestinal barrier ○ Prevents pathogens, toxins, and bacterial dislocation from gut to blood ○ Reduces colon damage ○ Suppresses colonic inflammation	[81]
Neurological disorders	Apigenin, CAPE, chrysin, kaempferol, pinocembrin, and quercetin	○ Alzheimer’s disease ○ Parkinson’s disease ○ Epilepsy ○ Ischemia	○ Inhibits the production of NO in microglia ○ Neuroprotection against Ischemia/reperfusion-induced injury ○ Prevent inflammatory stress ○ Blocks the activation of NF-κB in microglia ○ Neuroprotection against apoptosis and oxidation ○ Enhances memory impairment by inhibiting oxidative damage ○ Inhibits ROS formation	[24,82,83,84,85]
Respiratory tract-related diseases	Artepillin C, baccharin, CAPE, chrysin, galangin, kaempferide, kaempferol, naringenin, pinocembrin, benzyl caffeate, geranyl caffeate, and 3-methyl-2-butenyl caffeate	○ Asthma	○ Inhibits mast cell degranulation ○ Inhibits allergen-induced inflammation ○ Inhibits ROS production ○ Blocks NF-κB expression in macrophage cell lineage ○ Exhibits anti-inflammatory and anti-allergic activities	[85,86,87]
	Caffeic acid, CAPE, cinnamic acid, aromandendrin, N-acetylcysteine, p-coumaric acid,	○ Chronic obstructive pulmonary disease (COPD)	○ Prevents acute lung inflammation ○ Reduces stomatitis, oral infections, and dental plaque ○ Inhibits the NF-κB pathway ○ Reduces pro-inflammatory cytokines	[88,89,90,91]

### 3.4. Gastrointestinal Disorder

Gastrointestinal (GI) disorder is a medical condition that affects the motility of gastrointestinal tract, which includes a range of vital digestive organs such as the mouth, esophagus, stomach, bowels, rectum, and anus [92]. Propolis has been explored and emphasized in the treatment of GI disorder. Nowadays, a variety of GI disorders, including those that affect GI disorders, such as cancer, oral mucositis, peptic, gastritis, colitis, and ulcers of mucositis, have been successfully treated with propolis due to its bioactive components with various biological and therapeutic activities.

One of the common GI disorders is irritable bowel syndrome (IBS), which can be detected by stomach discomfort and altered bowel function [93]. It has been documented that the application of propolis via supplementation has improved the symptoms experienced by the IBS patients [93]. This might be due to the presence of quercetin glycosides in propolis, which reduce the visceral motor response, thus, increasing the pressure of pain threshold. Propolis has also been reported to inhibit the transcription of the inducible nitric oxide synthase (iNOS) gene, which can be influenced by NF-κB [93]. It has been documented that propolis reduces colon damage, suppresses colonic inflammation, and increases levels of mucosal and mucin secretion. Propolis can also improve the function of the intestinal barrier from the upregulation of the intestinal protein gene [94], as propolis is rich in prebiotics that improve the intestinal barriers, thus preventing the dislocations of bacterial, pathogens, and toxins from the gut to the blood stream [81,94].

### 3.5. Neurological Disorders

Neurological disorder is a medical condition that affects the human brain, especially the nerves that connect the spinal cord around the body. There are different types of neurological disorders, including Alzheimer’s disease, Parkinson’s disease, epilepsy, and ischemia, which are accompanied by the slow immune response of the brain, elevated oxidative stress, and induced inflammatory signaling [95]. The main mechanisms essential in cognitive impairment, neurodegeneration, and neurological disorder in the brain have been reported to be oxidative stress and neuroinflammations. Several studies have affirmed that propolis can be utilized in the treatment of these disorders because of the its potential anti-inflammatory, antioxidant, and immunomodulatory neuropharmacological effects of its active constituents [32]. This is because the lipophilic properties of propolis allow most of the major components of propolis to access the brain and navigate the functions of the blood-brain barrier [10,96].

Some of the metabolite classes of propolis compounds, such as alkaloids, saponins, sterols, tannins, and terpenoids, have positive effects on the central nervous system (CNS) [97]. Meanwhile, polyphenolic compounds such as kaempferol, pinocembrin, and quercetin are considered to be biomarkers in antioxidant activity that are beneficial for oxidative stress and have anti-inflammatory effects to modulate neurological disorders. These polyphenolic compounds have been proven to stimulate neurotransmitters, scavenge free radicals, and inhibit specific enzymes [98].

A systematic review study to assess the potential of propolis to protect the brain and treat neurological conditions and injuries was summarized by Zulhendri et al. [99]. It was suggested that propolis was repeatedly shown to decrease the expression of inflammatory and oxidative markers, such as TNF and nitric oxide, in organisms and cell cultures exposed to chemical and radiation toxicity, while increasing and maintaining antioxidant parameters, including superoxide dismutase. The presence of propolis prevented apoptosis by lowering the expression of the protein-coding genes which are linked to apoptosis signaling pathways. It is also believed that propolis helped to protect cell membranes and prevent tissue morphology from further deteriorating due to toxicity [99].

CAPE and pinocembrin also play an essential role in the treatment of these neurological diseases due to their neuroprotective properties [82,83]. Some of the aromatic acids of propolis including caffeic acid, CAPE, and vanillin have the potential to enhance memory loss due to their immunomodulatory, antioxidant, and anticancer properties. In terms of antioxidant activity, CAPE, which acts as selective inhibitor of the NF-κB activation, also decreases lipid peroxidation by inhibiting the cyclooxygenase-2 overproduction that is mediated by NF-κB transcription. Machado et al. reported that lupeol exhibits antidepressant and anticonvulsant activities that are important in the treatment of the CNS [84]. Meanwhile, pinocembrin has been described to elicit antioxidant and anti-inflammatory effects on the CNS by reacting on the mitogen-activated protein kinase signaling cascade and to modulate NF-κB, thereby preventing overstimulation [85]. Furthermore, the therapeutic potential of propolis appears to be based on its ability to reduce the expression of pro-inflammatory mediators and the generation of ROS, with it also stimulating neuroprotective factors and antioxidants [24].

### 3.6. Respiratory Tract-Related Disease

#### 3.6.1. Asthma

Allergic disorders have become a global health burden, resulting in a number of challenges due to the adverse factors of medicinal accessibility, high costs in terms of therapy, and a lack of therapeutic potential for patients. Nevertheless, therapeutic treatments of allergic reactions, such as asthma, atopic dermatitis, and allergic rhinitis have been well documented and supported by many preclinical and clinical studies. Asthma is a long-term inflammatory condition that causes narrow and swollen airways in the lungs, resulting in chest tightness, wheezing, and difficulty breathing, especially due to allergen inhalation. Research on asthma and its alternative medicinal treatments using natural resources including bee products such as propolis has substantially increased. Evidently, the active compounds of propolis have beneficial effects on asthmatic conditions due to its anti-allergic, anti-asthmatic, and anti-inflammatory properties, which could be attributed to the inhibitory effects on the activation of basophil and mast cells [100,101]. These two cells are considered to be the primary effector cells in allergic inflammation reactions [101]. Immunoglobulin E (IgE) receptors are cross-linked when an allergen binds to IgE, thus activating basophils and mast cells and resulting in the release of inflammatory mediators such as histamine and pro-inflammatory cytokines such as TNF-α [102]. The inhibitory mechanisms of propolis in allergic reactions can be seen in Figure 5.

It has been suggested that the aqueous extract of Taiwanese propolis consists of T helper cell response regulators that are beneficial in the context of asthmatic conditions due to their modulatory effects on both T helper cell responses [86]. These T cells can cause tissue remodeling and airway hyper responsiveness, which are associated with asthma [103]. Khayyal et al. highlighted that alternative medicines enriched with propolis have resulted in a decrease in the frequency and severity of the asthma attacks, which can be correlated with the decrease in inflammatory markers, such as TNF-α, IL-4, IL-5, and IL-6 [104]. The active components of propolis, including flavonoids compounds (chrysin, galangin, kaempferide, kaempferol, and pinocembrin), have been revealed to have substantial medicinal value, especially in in vivo studies of allergic disorders including asthma [86]. Nakamura et al. found that both CAPE and kaempferol have contributed to the anti-allergic activities of Chinese and Brazilian propolis extracts [100]. According to Tani et al., cinnamic acid derivatives such as artepillin C, baccharin, CAPE, kaempferide, and naringenin are also present in Brazilian propolis and show anti-allergic effects, and, in particular, the effects of artepillin C on allergen-induced allergic inflammation have been evaluated [105]. Notably, an in vitro study showed that active compounds of propolis including artepillin C, CAPE, benzyl caffeate, geranyl caffeate, and 3-methyl-2-butenyl caffeate are considered to be the primary contact allergen of propolis [106]. Another in vitro study demonstrated that artepillin C can inhibit cytokine secretions and ROS production, blocking the expression of NF-κB in macrophage cell lineage [87,88]. CAPE also inhibits the generation and activation of asthma by monocyte-derived dendritic cells [107]. Strong inflammatory flavonoids, such as quercetin, relieve asthma by suppressing exotoxin and IL-13, reducing eosinophilic mediators and type 2 helper cytokines. As a result, propolis and its active compounds will be able to effectively control and regulate asthma.

#### 3.6.2. Chronic Obstructive Pulmonary Disease

Chronic obstructive pulmonary disease (COPD) is a chronic lung disease that is caused by an obstruction of airflow from the lungs that results in multiple problems including breathing difficulties, wheezing, coughing, and mucus production [108]. There are several causes of COPD, including genetic disorders, air pollution, exposure to dust, and the most common is smoking [108]. Bronchodilators, corticosteroids, oxygen therapy, vaccines, and antibiotics are some of the optional treatments for COPD. The applications of natural products in antibiotics for COPD treatment have increased [109]. Many different types of natural products, such as ginger, thyme, curcumin, red sage, ginseng, and licorice roots have been used in multiple respiratory treatments [110]. Recently, numerous studies have reported that bee products, such as propolis, have useful and effective benefits in the treatment of asthma, respiratory diseases, and coughs [1,89,111]. This is because propolis contains numerous effective components such as flavonoids, phenolic compounds, phenolic acids, terpenes, terpenoids, not to mention its broad spectrum of therapeutic properties, such as anti-inflammatory, antioxidant, antimicrobial, and immunomodulatory properties [1,89,111,112,113,114]. Various clinical trials have been evaluated to validate the efficacy of propolis and its bioactive components as alternative complementary therapy in minimizing inflammatory-related conditions [104]. Inflammation was observed to have been reduced due to the synergistic effect of active compounds present in propolis [89,90,91]. Various clinical trials have been studied to validate the efficacy of propolis and its bioactive components as alternative complementary therapy in minimizing inflammatory-related conditions [104]. Inflammation reduces due to the synergistic effect of active compounds present in propolis [89,90,91].

Flavonoids, pinocembrin, polyphenols, alkaloids, coumarins, saponins, steroids, and triterpenoids have been reported to be prominent therapeutics against respiratory inflammation [115]. Patients with COPD have been reported to have reduced sputum production after using the propolis and N-acetylcysteine formulation [104]. de Moura et al. suggested that aqueous extract of Brazilian propolis exhibited anti-inflammatory and antiangiogenic due to the presence of CAPE, artepillin C, and caffeoylquinic in the extract [116]. Propolis has various biological targets and cellular effects as well as antioxidant, anti-inflammatory, and anticancer properties. Thus, it has the ability to reduce multiple pro-inflammatory cytokines, including TNF-α, IL-6, and IL-8.

A recent study documented that propolis successfully treated stomatitis and reduced oral infections as well as dental plaque, which can cause ulceration [117]. The antimicrobial properties from a mouthwash that contains propolis have helped to protect from oral infections [118]. Furthermore, by reducing pro-inflammatory cytokines as well as inducing apoptosis in macrophages, CAPE has the ability to inhibit the NF-κB pathway, resulting in the upregulation of NF-κB in macrophages and epithelial cells [119]. Flavonoids such as kaempferol, luteolin, and naringenin have been shown to exhibit activities that might be related to their antioxidant and anti-inflammatory properties.

## 4. Conclusions

In conclusion, propolis has served as a rich source in the context of drug discovery and has shown great potential for treatment in various acute and chronic human diseases. Research efforts on propolis have been intensively devoted in such frameworks. Propolis has been demonstrated to have some adverse effects on the human body. The potentially active phytochemical constituents of propolis include flavonoids, phenolic compounds, polyphenols, terpenes, terpenoids, coumarins, steroids, amino acids, and aromatic acids. The composition of phytochemical constituents varies according to bee species, geographical location, botanical source, and environmental conditions. Nevertheless, the biologically active compounds of propolis are artepillin C, caffeic acid, CAPE, apigenin, chrysin, galangin, kaempferol, luteolin, genistein, naringin, pinocembrin, coumaric acid, and quercetin, which have a broad spectrum of biological and therapeutic properties such as antidiabetic, anti-inflammatory, antioxidant, anticancer, cardioprotective, neuroprotective, immunomodulatory, and immuno-inflammatory agents. Propolis has been implemented in both clinical and physiological stages with various conditions. Most of the reported studies are based on experimental stage and randomized clinical trials, and, hence, further studies on the treatment of acute and chronic diseases with propolis would be of interest. It is also necessary to expand the wide application of propolis and to further explore its isolated bioactive components, which have multiple targets and various pharmacological activities. An improvement of the safety of propolis products, especially for oral supplementation, and its efficacy in long-term treatments, are also important to ensure its safe use.

## Figures and Tables

**Figure 1 molecules-27-06120-f001:**
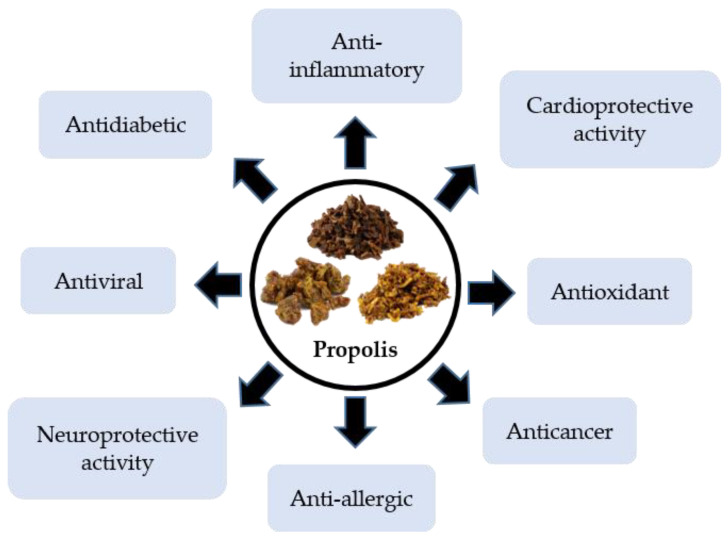
Biological and therapeutic properties of propolis.

**Figure 2 molecules-27-06120-f002:**
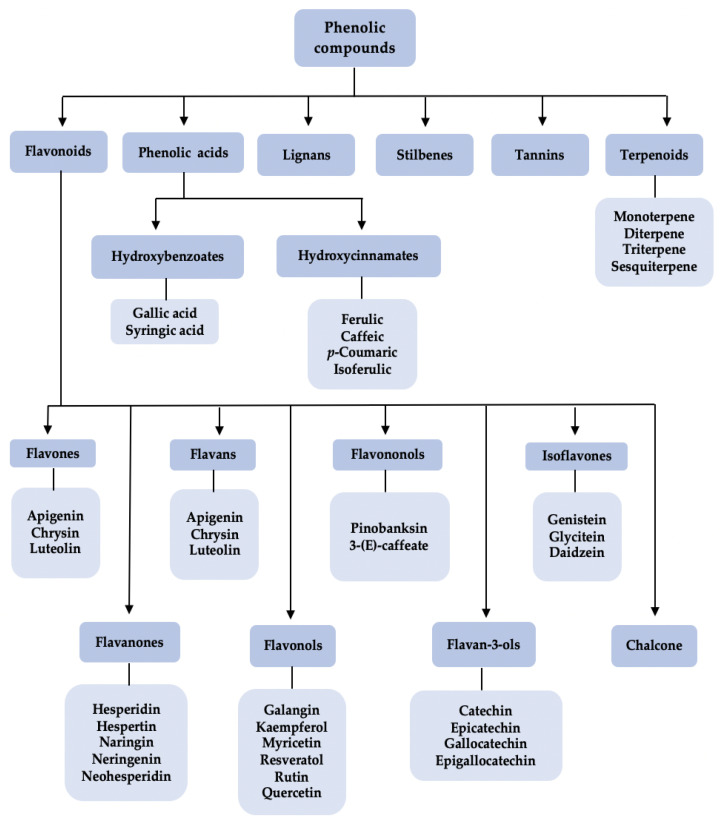
Classification of phenolic compounds.

**Figure 3 molecules-27-06120-f003:**
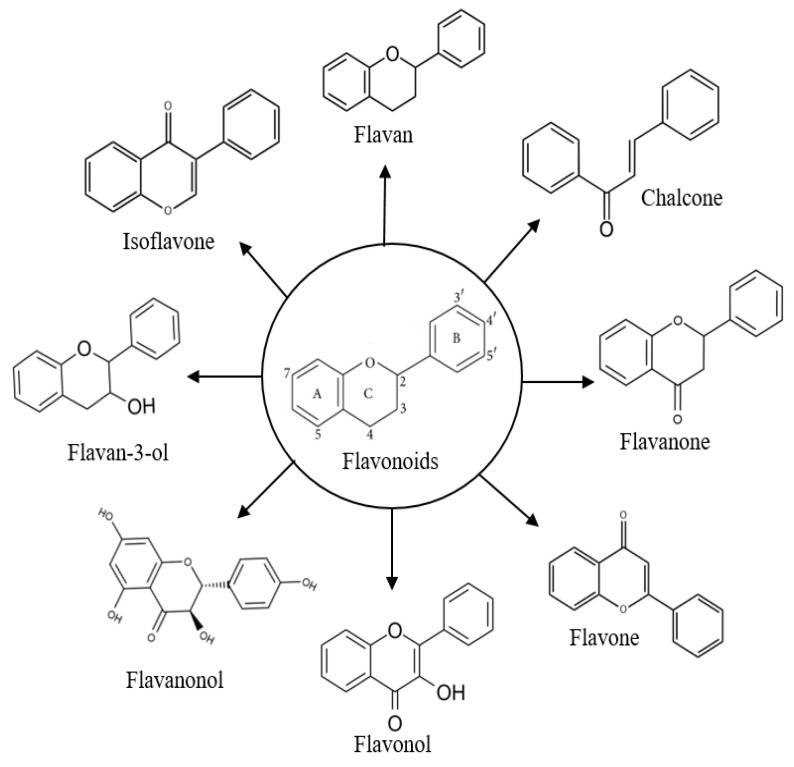
Chemical structures of sub-groups of flavonoids, showing that the various sub-groups are related to the oxidation and substitution of the heterocyclic C ring, while individual compounds in a sub-group vary depending on the substituents attached to the A and B rings.

**Figure 4 molecules-27-06120-f004:**
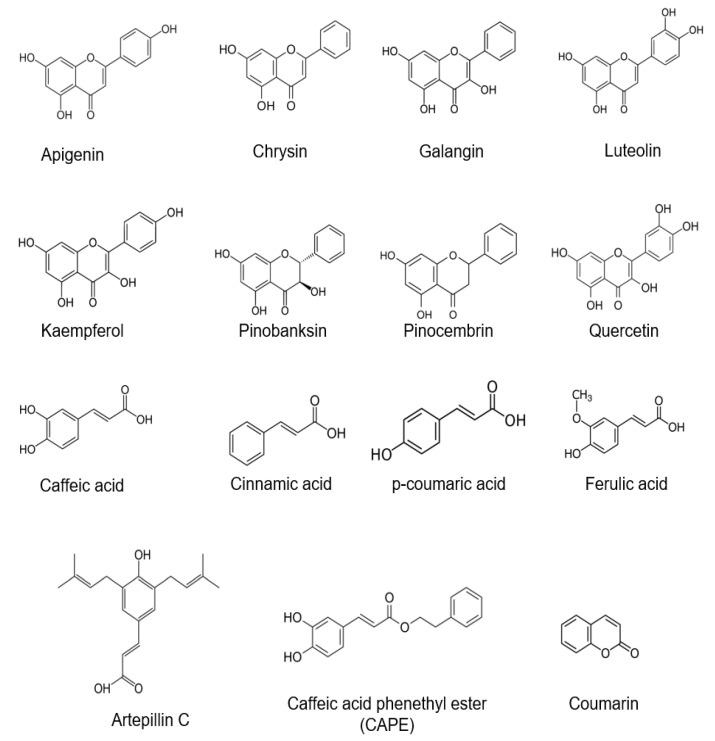
Common bioactive compounds in propolis.

**Figure 5 molecules-27-06120-f005:**
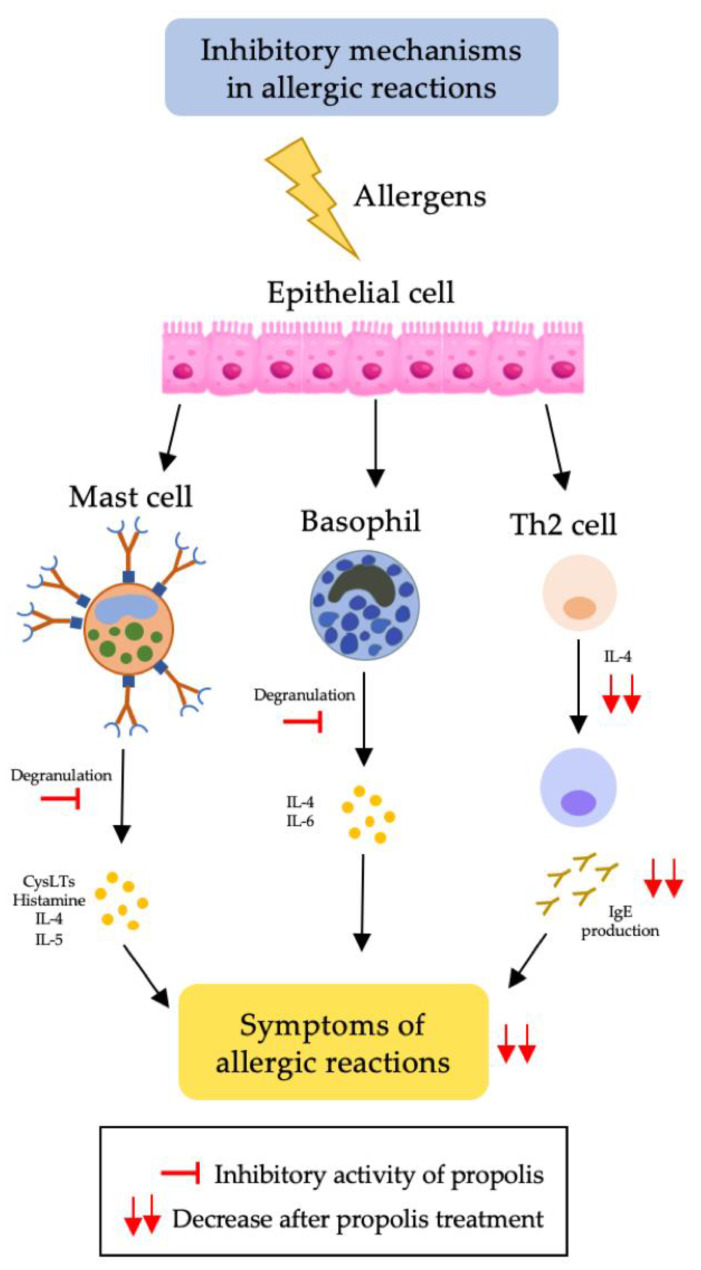
Inhibitory mechanisms of propolis in allergic reactions.

**Table 1 molecules-27-06120-t001:** Chemical compositions of propolis and their bioactive compounds.

Chemical Compositions	Bioactive Compounds	Ref.
Aromatic acids	Benzoic acids, caffeic acid, cinnamic acid, coumaric acid, ferulic acid, and gallic acid	[12,15,22,23,24,25]
Alcohols	Glycerol, erythritol, α-cedrol, xylitol	
Esters	Caffeic acid phenethyl ester, 2-propenoic acid methyl ester, 4,3-acetyloxycaffeate, 3,4 dimethoxy-trimethylsilyl esters, and 3-methoxy-4-cinnamate	
Fatty and aliphatic acids	Isoferulic acid, glutamic acid, phosphoric acid, malic acid, tartaric acid, propanoic acid, butanedoic acid, and stearic acid	
Flavonoids	Apigenin, acacetin, chrysin, galangin, genistein, hesperetin, kaempferol, kaempferide, luteolin, naringenin, pinobanksin, pinocembrin, quercetin, and tetrochrysin	
Microelements	Aluminium (Al), copper (Cu), magnesium (Mg), zinc (Zn), silicon (Si), tin (Sn), manganese (Mn), nickel (Ni), and chrome (Cr)	
Sugars	d-Altrose, d-glucose, maltose, and d-fructose	
Vitamins	Vitamin A (retinol), vitamin B_1_ (thiamine), vitamin B_2_ (riboflavin), vitamin B_3_ (nicotinamide), vitamin B_6_ (pyridoxine), vitamin B_9_ (folic acid), vitamin C (ascorbic acid), and vitamin E (tocopherol)	
Others	Butane, cyclohexane, cyclopentene, and guanidine	

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
