# Peer review of "Propolis: Its Role and Efficacy in Human Health and Diseases"

_molecules, 2022, doi:10.3390/molecules27186120_

Round 1
Reviewer 1 Report
See Abstract, line 10: The authors intend to focus on stingless bee propolis. Perhaps, the title needs to be modified to reflect that.
Line 38: please add info about Apis mellifera as well. Propolis is not only produced by stingless bees.
Line 90-95, Table 1: The objective of the paper is to describe the potential of stingless bee propolis. The authors need to modify and add emphasis on the bioactive compounds of stingless bee propolis. The references appear to include Apis mellifera propolis as well.
Similar problem can be observed across the entire manuscript.
The term "scheme" can be changed to the more standardized term: "figure".
Diseases:
1. Rheumatoid arthritis: recent clinical study by Matsumoto et al. 2021 PLoS ONE 16(5): e0252357 needs to be included and discussed.
2. Anticancer: Human cinical studies such as J Radiat Res Appl Sci. 2016;9:431-440, Eur J Cancer Care. 2017;26:e12757, J Herb Med. 2020;23:100385, need to be properly included and discussed.
3. Coronavirus disease: Clinical trial by Silveira et al. 2021. Biomedicine and Pharmacotherapy Volume 138, June 2021, 111526. needs to be included
4. Neurological disorder: Human clinical trials re this have also been done. See review by Zulhendri et al. 2021. Biomedicines 9(9): 1227.
The authors need to include all available human clinical trials in the discussion re the use of propolis in these diseases.
General comments:
1. The authors put emphasis on stingless bee propolis in the abstract and introduction. However in the body of the manuscript, the authors are jumping around between Apis mellifera and stingless bee propolis. Moreover, detailed discussion re stingless bee propolis is also lacking.
Therefore, the manuscript appears to be "confused" and is not focused enough. Perhaps the authors can modify the abstract and introduction - remove the term stingless bees, and/or add more discussion re stingless bee propolis.
2. The manuscript appears to be redundant. The present review manuscript repeats many details that have been covered by other recent published review papers without any significant contribution.
3. Perhaps, the authors can add more details re the proposed mechanisms of action of propolis (that have not been covered by other recent published papers) with regards to the diseases discussed in the present manuscript. Not just merely repeating other present review papers.
In the present form, unfortunately, the manuscript lacks rigor.
Author Response
Reviewer #1
Comments and recommendations
- See Abstract, line 10: The authors intend to focus on stingless bee propolis. Perhaps, the title needs to be modified to reflect that.
Our response: We thank the reviewer for her/his sincere comment and suggestion. We have included the discussion and reports on the uses of both Apis mellifera and stingless bees propolis in human health and diseases, and we have compared them at some points. Accordingly, to clarify this issue, we have heavily revised the Abstract, Introduction, and throughout the manuscript, but we keep the title as it is.
- Line 38: please add info about Apis mellifera as well. Propolis is not only produced by stingless bees.
Our response: We thank the reviewer for her/his sincere comment and suggestion. We have added information on Apis mellifera in the paragraph.
- Line 90-95, Table 1: The objective of the paper is to describe the potential of stingless bee propolis. The authors need to modify and add emphasis on the bioactive compounds of stingless bee propolis. The references appear to include Apis mellifera propolis as well. Delete the references on Apis. Similar problem can be observed across the entire manuscript.
Our response: We thank the reviewer for her/his sincere comment and suggestion. As this review article covers both Apis mellifera and stingless bees propolis, we would like to keep the references on Apis mellifera propolis. We also appreciate reviewer’s comment; thus we have revised the last paragraph of Introduction.
- The term "scheme" can be changed to the more standardized term: "figure".
Our response: We have already changed “Scheme” to “Figure” as suggested.
- Diseases:
Rheumatoid arthritis: recent clinical study by Matsumoto et al. 2021 PLoS ONE 16(5): e0252357 needs to be included and discussed.
Anticancer: Human clinical studies such as J Radiat Res Appl Sci. 2016;9:431-440, Eur J Cancer Care. 2017;26:e12757, J Herb Med. 2020;23:100385, need to be properly included and discussed.
Coronavirus disease: Clinical trial by Silveira et al. 2021. Biomedicine and Pharmacotherapy Volume 138, June 2021, 111526. needs to be included
Neurological disorder: Human clinical trials re this have also been done. See review by Zulhendri et al. 2021. Biomedicines 9(9): 1227.
The authors need to include all available human clinical trials in the discussion re the use of propolis in these diseases.
Our response: We really thank the reviewer for her/his sincere suggestion. We have added the discussion on clinical trials of propolis in the treatments of Rheumatoid arthritis, cancer, virus, and neurological disorder. Accordingly, we have included all those suggested relevant papers in this revised manuscript.
- The authors put emphasis on stingless bee propolis in the abstract and introduction. However, in the body of the manuscript, the authors are jumping around between Apis mellifera and stingless bee propolis. Moreover, detailed discussion re stingless bee propolis is also lacking.
Therefore, the manuscript appears to be "confused" and is not focused enough. Perhaps the authors can modify the abstract and introduction - remove the term stingless bees, and/or add more discussion re stingless bee propolis.
Our response: We really thank the reviewer for her/his sincere comment and suggestion. We accept that this is confusing, as most of studies reported in literature have been focused more on A. mellifera propolis than stingless bee propolis. Therefore, as reviewer’s suggestion, we have included the discussion and reports on the uses of both Apis mellifera and stingless bees propolis in human health and diseases in the text.
- The manuscript appears to be redundant. The present review manuscript repeats many details that have been covered by other recent published review papers without any significant contribution.
Our response: We really understand the reviewer’s sincere comment that the review articles summarizing the health benefits of A. mellifera propolis and stingless bee propolis have been reported by many different research groups. However, there are several differences, particularly the scope, viewpoint, and perspective, between our paper and those published review papers. As the review papers summarize the findings based on the similar published papers in literature, it is unavoidable that some affirmative overlap should exist between our paper and those published review papers.
- Perhaps, the authors can add more details re the proposed mechanisms of action of propolis (that have not been covered by other recent published papers) with regards to the diseases discussed in the present manuscript. Not just merely repeating other present review papers.
In the present form, unfortunately, the manuscript lacks rigor.
Our response: We thank the reviewer for her/his sincere comment and suggestion. This is normal when researchers always try to understand the mechanisms of action of pharmaceutical compounds of propolis in in human health and diseases. However, most of them are still open questions. It would need multiple intensive in vivo, ex vivo and in vitro studies to confirm such mechanisms of actions. Therefore, we limited our summary to those that have been established. For such purpose, our ongoing research works are mainly focused on the in vitro studies of mechanisms of actions of propolis based on chemical constituents, phytochemicals, and antimicrobial studies.

Reviewer 2 Report
This article is well-written review on the recent studies on the role and efficacy of propolis in the medicinal and pharmaceutical field. However, the diversity of propolis depending on the geographical and plant origins is not described. Most description in this review is concerning with poplar origin propolis. The difference of the constituents in propolis depending on the source plants is important. Thus, the authors should add the description on the relationship between the constituents depending on plant origins and pharmaceutical activities.
Author Response
Reviewer #2
Comments and recommendations
This article is well-written review on the recent studies on the role and efficacy of propolis in the medicinal and pharmaceutical field. However, the diversity of propolis depending on the geographical and plant origins is not described. Most description in this review is concerning with poplar origin propolis. The difference of the constituents in propolis depending on the source plants is important. Thus, the authors should add the description on the relationship between the constituents depending on plant origins and pharmaceutical activities.
Our response: We thank the reviewer for her/his sincere encouragement and positive comments and suggestions. Accordingly, we have included all of the suggestions in this revised manuscript.
Round 2
Reviewer 1 Report
Line 445: Zulhendri et al. is a review paper. For clarity purposes, please amend " a clinical study" to " a review study".
Author Response
Line 445: Zulhendri et al. is a review paper. For clarity purposes, please amend " a clinical study" to " a review study".
Our response: We thank the reviewer for her/his correction. Accordingly, we have corrected the statement in line 445 to be; “A systematic review study to assess the potential of propolis to protect the brain and treat neurological conditions and injuries has been summarized by Zulhendri et al.”